# Bile Acids, Intestinal Barrier Dysfunction, and Related Diseases

**DOI:** 10.3390/cells12141888

**Published:** 2023-07-19

**Authors:** Linsen Shi, Lihua Jin, Wendong Huang

**Affiliations:** 1Department of Diabetes Complications and Metabolism, Arthur Riggs Diabetes and Metabolism Research Institute, Beckman Research Institute, City of Hope National Medical Center, 1500 E. Duarte Road, Duarte, CA 91010, USA; lshi@coh.org; 2Irell & Manella Graduate School of Biomedical Science, City of Hope National Medical Center, 1500 E. Duarte Road, Duarte, CA 91010, USA

**Keywords:** bile acid, bile acid receptor, gut microbiota, intestinal barrier, IBD, sepsis, NASH, CRC, aging

## Abstract

The intestinal barrier is a precisely regulated semi-permeable physiological structure that absorbs nutrients and protects the internal environment from infiltration of pathological molecules and microorganisms. Bile acids are small molecules synthesized from cholesterol in the liver, secreted into the duodenum, and transformed to secondary or tertiary bile acids by the gut microbiota. Bile acids interact with bile acid receptors (BARs) or gut microbiota, which plays a key role in maintaining the homeostasis of the intestinal barrier. In this review, we summarize and discuss the recent studies on bile acid disorder associated with intestinal barrier dysfunction and related diseases. We focus on the roles of bile acids, BARs, and gut microbiota in triggering intestinal barrier dysfunction. Insights for the future prevention and treatment of intestinal barrier dysfunction and related diseases are provided.

## 1. Introduction

The intestinal barrier is a highly complex and precisely controlled physiological structure. It interacts with the external environment as a physical, biochemical, and immunological barrier and regulates many critical homeostatic functions [1]. In health, the intestinal barrier is semi-permeable and protects the internal environment from the potential infiltration of pathological molecules and microorganisms while allowing for the absorption of nutrients and water [2]. However, under pathological situations, such as ischemia, trauma, stress, and infection, the integrity of the intestinal barrier is disrupted, leading to many local and systemic diseases.

Bile acids are hydroxylated sterols derived from cholesterol in the liver via either the classical (neutral) or alternative (acidic) pathway. Primary bile acids are synthesized in the liver, stored in the gallbladder, and then secreted into the intestine. In the gut, bile acids are transformed into secondary or tertiary bile acids by the microbiota [3]. Traditionally, bile acids were thought to be involved in the absorption and metabolism of lipid and lipid soluble vitamins. Recent data suggest that bile acids act as hormones and engage bile acid receptors (BARs), including farnesoid X receptor (FXR), Takeda G protein-coupled receptor 5 (TGR5), sphingosine-1-phosphate receptor 2 (S1PR2), pregnane X receptor (PXR), vitamin D receptor (VDR), and constitutive androstane receptor (CAR), to play important roles in the metabolism, inflammation, immune homeostasis, tumorigenesis, aging, and other aspects of organism [4,5,6,7,8,9].

There are many dynamic interactions and two-way cross talks between bile acids and the intestinal barrier. Bile acids are recognized as key molecules that control the integrity of the compromised intestinal barrier. In this review, we focus on the role of bile acids in the maintenance of intestinal barrier, the relationship between bile acid disorders and intestinal barrier dysfunction, and their related diseases. Additionally, we discuss the potential of modifying the metabolism of bile acids and their signaling pathways as therapeutic approaches to liver and intestinal disease.

## 2. Bile Acids and Intestinal Barrier

The integrity of the intestinal barrier requires constant renewal of the epithelial boundary, maintenance of tight junctions (physical barrier), mucus secretion and normal gut microbiota (biochemical barrier), and a finely regulated intestinal lamina propria immune system (immune barrier). Because of the complexity and heterogeneity of the intestinal barrier, specific mechanisms underlying the dysfunction of the intestinal barrier are still far from clear. That bile acids may play a pivotal role in many aspects of the maintenance of intestinal barrier integrity is a novel concept (Figure 1). Of interest, it is now understood that bile acids can activate specific BARs or other downstream signaling pathways to modulate biological functions of the intestinal mucosal barrier [10,11,12].

### 2.1. Bile Acids and Intestinal Epithelial Cells Tight Junctions

Tight junctions provide the main connection between intestinal epithelial cells and are formed by zonula occludens (ZOs), claudins (Cldns), and occludin (Ocln) proteins, which play an important role in maintaining the normal physiological function of epithelial cells [13]. The composition of bile acids is affected by diet, exercise, drugs, age, and other factors, and responds dynamically to local and whole-body ques (Figure 2). Alteration of the bile acid profile changes the permeability of the intestinal mucosa and affects the barrier function through regulating the expression of tight junction proteins. For example, a high fat diet (HFD)-induced increase in deoxycholic acid (DCA) is a major environmental factor in the development of colorectal cancer (CRC). Apart from inducing chronic inflammation, reductions in zonula occludens 1 (ZO-1) and goblet and Paneth cells were observed in *Apc*^min/+^ mice after DCA treatment [14]. In contrast, lithocholic acid (LCA) ameliorated the TNF-α-induced distribution of ZO-1, E-cadherin, occludin, and claudin-1 [15]. The administration of curcumin, a polyphenolic compound isolated from turmeric, decreased the lipopolysaccharides (LPS)-induced injury of intestinal tight junctions and alleviated acute inflammation in the mucosa, likely through altering the microbiome and modulation of bile acid metabolism [16].

Epithelial myosin light chain kinase (MLCK) was found to regulate tight junction protein expression and intestinal barrier function [17]. Related to this, chenodeoxycholic acid (CDCA) reversed an LPS-induced decrease in tight junction protein expression through activating MLCK [18]. Moreover, tauroursodeoxycholic acid (TUDCA), a bile acid commonly used for hepatobiliary diseases treatment, acting via TGR5-MLCK, actually improved the impairment of the *E. coli*-induced epithelial barrier [19].

### 2.2. Bile Acids and Gut Microbiota

The intestine hosts a co-evolved microbial ecosystem that is part of the intestinal mucosal barrier. The relationship between the gut microbiota and metabolism of bile acids was well reviewed [20,21,22] and not germane to the present focus. Here we focus on the bidirectional interactive feedback between bile acids and gut microbiota and its effects on the intestinal mucosal barrier function.

Ursodeoxycholic acid (UDCA) supplementation attenuated inflammation and reduced intestinal permeability caused by multidrug-resistant extended-spectrum β-lactamase (ESBL)-producing *E. coli* in colibacillus diarrhea. UDCA supplementation inhibited bacterial growth and invasion, alleviated commensal bacterial dysbiosis, and corrected colitis via the TGR5- nuclear factor-kappa B (NF-κB) pathway [23]. Fibroblast growth factor FGF 15/19 (FGF15/19) is mainly expressed in the intestine under the control of the FXR. The activation of the FXR-FGF19 axis modulated intestinal flora and inhibited intestinal inflammation via restoring the normal bile acid pool [24].

As gut microbiota constantly comes into contact with the external environment, the composition and function of the microbiota is susceptible to many factors, such as diet, medications, exercise, and emotions. In keeping with this, the consumption of an HFD modified the gut microbiome and bile acid pool and increased intestinal mucosal permeability [25]. In mice, dietary fiber with insulin altered the composition of the microbiota and the levels of microbiota-derived metabolites, notably bile acids, that triggered type 2 inflammation at barrier surfaces [10,26]. Some special diets, such as methionine-restricted diets (MRDs) and l-Glutamine, enhanced the intestinal barrier integrity by regulating the intestinal microbiota and bile acids profiles [27,28]. And a BAR signaling-independent, physicochemical mechanism for conjugated the BA-mediated protection of epithelial barrier function was described. In this situation, conjugated bile acids, through micelle formation, protected the intestinal epithelium from damage by unconjugated bile acids [29] (Figure 1).

### 2.3. Bile Acids, Intestinal Stem Cells (ISCs), and Epithelial Injury Repair

Balance between intestinal epithelial proliferation and cell death from damage, stress, and other pathological conditions maintain normal intestinal barrier function. Intestinal epithelial cells are self-renewed every 3–5 days, in part, from Lgr5+ intestinal stem cell (ISCs) as at counteract intestinal barrier damage and different stress stimuli [30]. Lgr5+ ISCs replenish damaged epithelial cells and generate progenitors of goblet and Paneth cells. These cells secrete mucus and antimicrobial peptides to support the integrity of the intestinal mucus layer [31].

Data demonstrated that bile acids metabolism plays a potential role in the self-renewal function of Lgr5+ ISCs. Based on pathway synthesis and chemical structures, bile acids are grouped into 12α-hydroxylated (OH) bile acids and non-12α-OH bile acids [3]. What the two groups exert varies, and, at times, they have opposing effects on ISCs. For instance, the 12α-OH bile acid CA can inhibit the activity of peroxisome proliferator-activated receptor alpha (PPARα), impeding fatty acid oxidation (FAO), and the self-renewal of Lgr5+ ISCs [32]. An HFD-driven increase in DCA decreased ISC proliferation and differentiate into goblet cells through pathologic endoplasmic reticulum stress [33]. In contrast, the non-12α-OH bile acid LCA activates TGR5 and downstream proto-oncogene tyrosine-protein kinase (SRC) and Yes-associated protein (YAP) pathways to promote ISC renewal [34]. The 12α-OH bile acid deoxycholic acid (DCA) inhibited mucosal healing in mice, but the non-12α-OH bile acid UDCA inhibited FXR activity and increased the expression of the cystic fibrosis transmembrane conductance regulator (CFTR) Cl channels in colonic epithelial cells to promote mucosal healing [35]. Dcha-20, a novel LCA derivative with vitamin-D-like activity, upregulated the expression and activity of CYP3A4, an indicator of intestinal functional maturation, in a human-induced pluripotent stem-cell-derived intestinal organoid [34]. This suggests a novel strategy to enhance the regenerative capacity of the intestinal epithelium and promote epithelial injury repair [36].

### 2.4. Bile Acids and Intestinal Local Immune Homeostasis

The intestinal lamina propria is colonized by a variety of innate and adaptive immune cells and gut-associated lymphoid tissue and is termed the intestinal immune barrier [37,38,39]. Under normal conditions, this microecosystem is tightly and finely regulated. Environmental factors and the gut microbiota and their metabolites (microorganism-associated molecular patterns and pathogen-associated molecular patterns) are recognized by specific receptors (Toll-like receptors, TLRs) on immune cells, leading to intestinal immune homeostasis and self-tolerance [39].

Bile acids modulate immune responses in the intestine through BARs, including TGR5, FXR, VDR, CAR, and retinoic-acid-receptor-related orphan nuclear receptor-γt (RORγt). Unique lymphocyte populations function cooperatively to maintain the intestinal immune system, especially the balance between pro-inflammatory T helper 17 (Th17) cells and anti-inflammatory Treg cells [40]. Two derivatives of LCA, 3-oxoLCA and isoalloLCA, inhibited RORγt to suppress Th17 cell differentiation and increased the differentiation of Treg cells through the production of mitochondrial reactive oxygen species [41]. The secondary bile acid, isoLCA, via VDR, modulated colonic FOXP3 + Treg cells expressing RORγt and ameliorated their susceptibility to inflammatory colitis [42]. The secondary bile acid, 3β-hydroxydeoxycholic acid (isoDCA), increased Foxp3 and suppressed dendritic cell immunostimulatory properties [43].

In addition to T lymphocytes, bile acids can also affect the homeostasis of the intestinal immune barrier through modifying the function of macrophages. HFD leads to systemic low-grade inflammation in the intestinal mucosa. This effect was related to changes in gut microbiota and bile acids (e.g., increased CA and DCA) and, subsequently, M1 macrophage polarization and pro-inflammatory cytokines production [25,44]. In macrophages, UDCA-FXR signaling suppressed NF-κB activation and reduced inflammatory cytokine production, promoting M2 macrophage polarization in low-birth-weight piglets [45]. Furthermore, systemic FXR activation lowered bile acid synthesis, suppressed macrophages production of IL1β and TNFα, and inhibited Th1/Th17 lymphocyte polarization [46].

## 3. Bile Acid and Intestinal-Barrier-Dysfunction-Related Diseases

Bile acids and BARs are involved in different types of diseases and disorders, including diabetes, obesity, fatty liver, cardiovascular disease, lung disease, and cancer. In this section, we focus on the diseases related to the disfunction of the intestinal barrier caused by bile acid metabolic dysregulation (Figure 2) (Table 1).

### 3.1. Inflammatory Bowel Diseases (IBDs)

Inflammatory bowel disease (IBD) is a chronic non-specific inflammation of the gastrointestinal tract. IBD most often manifests as ulcerative colitis (UC) or Crohn’s disease (CD) [74]. Apart from local immune dysregulation and auto inflammation in the intestinal mucosa, increased intestinal permeability is associated with disease progression [75]. Bile acids and gut microbiota contribute to mucosal barrier integrity and homeostasis. Not surprisingly, the mutual interaction between them is undisputedly related to the pathogenesis, prevention, and therapy of IBD [12,21,76,77]. The dysbiosis of bile acids and bile acid signaling participate in the occurrence and progression of IBD. Indeed, altered bile acid profiles were reported in IBD [12,47,48,49,77]. Increased primary bile acids and decreased secondary bile acids, particularly DCA and LCA, were characteristic of active IBD [47,48,49]. Individuals with Crohn’s disease, but not ulcerative colitis, had a reduced bile acid pool size compared to individuals without IBD [78]. Further, the microbial gene pathways involved in secondary bile acid biosynthesis were found to be depleted in the terminal ileum of individuals with IBD patients compared with healthy controls [79].

Some bile acids altered the expression of tight junction proteins and the renewal of intestinal stem cells, leading to the intestinal barrier injury, increasing the incidence of IBD [32,50]. In contrast, some other bile acids may maintain intestinal immune barrier homeostasis by activating BARs such as FXR and TGR5. The DCA-mediated activation of FXR inhibited Paneth cell function and type I interferon signaling in mice with Crohn’s disease [51]. Nuclear xenobiotic receptor CAR signaling altered the transcriptome of Teff cells that infiltrated the small intestine lamina propria (siLP) and suppressed Crohn’s disease-like small bowel inflammation [80]. Relevant to IBD, an association between cooperation within the gut microbiota, such as the generation of LCA or metabolite butyrate, and the modulation of P-glycoprotein (P-gp), was demonstrated [81].

Immunosuppressants, glucocorticoids, amino salicylic acid, and tumor necrosis factor antagonists are employed in the treatment of IBD. However, these agents have unwanted side effects and only modest efficacy. Bile acids and their derivatives, and BAR regulators, are treatment strategies for IBD. In line with this, secondary bile acids, such as UDCA- and LCA-induced activation of TGR5, improved gut barrier integrity and reduced the inflammation in murine colitis [82,83]. Until now, most of the BAR-targeting drugs developed for IBD focused on the agonists of FXR and TGR5 [83,84,85].

### 3.2. Gut Origin of Sepsis

Sepsis is a serious clinical syndrome in critically ill patients caused by systemic infection and abnormally activated immune response [86]. Extending this, the gut is thought to be a “motor” of sepsis and related multiple organ failure (MOF) [87]. In this capacity, intestinal barrier dysfunction and bacterial translocation (BT) would worsen any infectious process [88]. Gut-origin sepsis involves bacteria and bacteria-associated products crossing a disrupted intestinal mucosal barrier into the mesenteric lymph nodes and their circulation [89].

The role of bile acids in intestinal barrier retention is well confirmed. However, there is also a close relationship between the metabolism of bile acids and gut-originating sepsis. Consistent with this view, the inflammatory mediators released during sepsis inhibited hepatobiliary transporter gene expression, resulting in hyperbilirubinemia and cholestasis [90]. Serum bile acid concentrations were significantly higher in animals and humans with sepsis. Thus, bile acids may be a potential marker for early sepsis [52,53]. Of interest is the analysis of plasma from septic individuals found to be glycochenodeoxycholate- and phenylalanine-associated with survival of sepsis [91].

Strategies targeting bile acids and bile acid pathways may be a treatment for gut-origin sepsis. TUDCA, a hydrophilic bile acid used for the treatment of various cholestatic disorders, stimulated intestinal epithelial cell migration and preserved the intestinal barrier. Acting through TGR5-NF-κB, it also prevented sepsis-mediated cholestasis and bacterial dysbiosis [23,54,55]. *Burkholderia pseudomallei* is responsible for up to 40% sepsis-related mortality. TUDCA promoted *B. pseudomallei* clearance by inhibiting endoplasmic reticulum (ER) stress-induced apoptosis [56]. Recently, conjugated bile acids have been found to increase the integrity of the gut barrier [29,57]. Simple oral gavage with conjugated bile acids decreased bacterial translocation and endotoxemia and increased survival in septic mice [92]. Another bile acid derivative, camptothecins bile acid, inhibited NF-κB and alleviated sepsis-induced liver injury [93].

Related to this, the probiotic *Lactobacillus rhamnosus* reduced sepsis mortality by rebalancing metabolic bile acid profiles [94]. Babaodan, a natural preparation, appeared to alter NF-κB and NLRP3 (NLR family pyrin domain containing 3) inflammasome complex assembly to limit LPS-induced sepsis [95]. The beneficial actions of Babaodan were believed to be due, in part, to bile acids. FGF19 inhibited bile acid synthesis by suppressing CYP7A1. In sceptic mice, pretreatment with FGF19 was protective against LPS-induced liver, ileum, and kidney injury [96]. Activation of bile acid receptors FXR and TGR5 altered the NLRP3 inflammasome and cAMP/PKA/CREB signaling to decrease sepsis [97,98].

### 3.3. Non-Alcoholic Fatty Liver Disease (NAFLD)

Non-alcoholic fatty liver disease (NAFLD) is a common chronic liver disease with a spectrum of severity, including non-alcoholic fatty liver disease, non-alcoholic steatohepatitis (NASH), cirrhosis, and secondary hepatocellular carcinoma [99,100,101]. Although the pathological mechanism of NAFLD has not been fully elucidated, increased intestinal permeability and impaired intestinal barrier function participate in the disease [102,103,104]. Because about 70–75% of the liver blood supply comes from the portal vein, which drains blood from the mesenteric veins of the intestinal tract, the liver is, in the face of disrupted barrier function, the first-line organ to encounter the endotoxin and bacterial components translocated from intestine [101].

Bile acids may also affect the progression of NAFLD through adjusting the intestinal barrier [105,106,107]. Long-term HFD intake can lead to gut dysbiosis and the aberrant metabolism of bile acids. Bidirectional crosstalk between gut microbiota and bile acids could impair the intestinal epithelial function, increasing the translocation of gut-derived endotoxins such as LPS to the blood and lymphatics to activate hepatic TLR-4/NF-κB signaling and promote NAFLD or NASH [108,109]. HFD-mediated changes in bile acids decreased FXR and TGR5 signaling and degraded the intestinal barrier [110,111].

Due to its insidious onset, NAFLD is difficult to diagnose at an early stage. Alterations in bile acid homeostasis were associated with NASH and liver fibrosis. Therefore, bile acids may be promising non-invasive diagnostic biomarkers for NAFLD [112,113]. Circulating levels of DCA and gut microbiota containing DCA generating genes increased with NAFLD severity and fibrosis stage [58,59]. Yet, no differences in total bile acids were seen between NAFLD and NASH. Closer inspection did note that primary conjugated bile acids increased, and unconjugated bile acids significantly decreased in relation to the degree of liver fibrosis [60].

OCA, a CDCA derivative and FXR agonist, interfered with TLR4/TGF-β1 signaling to activate autophagy and intestinal integrity in NASH [61,62,63]. TUDCA attenuated the progression of HFD-induced NAFLD in mice and was associated with less gut inflammation, better intestinal barrier function, and changes in the microbiota composition [64]. Furthermore, natural compounds from plants and exercise were found to lessen NAFLD, in part, by regulating bile acid metabolism, counteracting HFD-induced microbial imbalance, and supporting the intestinal barrier [114,115,116].

### 3.4. Colorectal Cancer (CRC)

CRC is one of the most prevalent cancers worldwide and is linked to environmental factors, particular HFD [117,118,119]. Long-term HFD caused dysbiosis and a shift in the bile acids profile, especially unconjugated bile acids and secondary bile acids [65,66]. The dysbiosis of bile acids, such as high levels of CA and DCA and an increased 12α-OH/non-12α-OH bile acids ratio, promoted dysregulated immunity, loss of the intestinal barrier, invasion of microbial and pathogenic metabolites, and increased inflammation, all of which could increase CRC [67].

Elevated levels of fecal secondary bile acids, especially DCA, were associated with an increased risk of CRC [120,121,122]. Oral treatment with DCA in *Apc*^min/+^ mice reduced the expression of tight junction proteins and the number of intestinal goblet and Paneth cells, induced low-grade inflammation, and aggravated intestinal tumorigenesis [14,123]. HFD-mediated changes in the gut microbiome and increased secondary bile acids invoked Wnt signaling with epithelial cell proliferation and colonic neoplasm [124]. Likewise, gavage with CA for 10 weeks markedly increased intestinal adenoma progression along with impaired intestinal barrier function and IL-6/STAT3-related low-grade inflammation [125].

The dysregulation of bile acids-BAR pathways also plays a pivotal role in the progression of CRC. Decreased FXR-FGF15 signaling and overexpressed TGR5 were observed in azoxymethane (AOM)/dextran sodium sulfate (DSS)-induced CAC mice [68]. In contrast, the activation of FXR protected the intestinal barrier, decreased inflammation, and restricted tumor growth [126]. HFD and dysregulated Wnt signaling (APC mutation) altered bile acids profiles, increased the malignant transformation of Lgr5+ cancer stem cells, and promoted adenocarcinoma progression which was counteracted by the activation of FXR [127]. Additionally, kaempferol upregulated FXR expression and increased CDCA to decrease tumor growth in *Apc*^min/+^ mice [128]. OCA treatment enhanced FXR binding to the suppressor of the cytokine signaling 3 (SOCS3) promoter, increased SOCS3, and decreased Janus kinase 2 (JAK2)/signal transducer and activator of transcription 3 (STAT3) signaling to limit tumorigenesis [129]. Furthermore, an FXR agonist plus GSK126 (an EZH2 inhibitor) showed synergistic anti-tumor effects [130].

### 3.5. Aging 

Aging is defined as a progressive decline in cellular and organismal function [131]. Aging is associated with genomic instability, epigenetic alterations, telomere attrition, loss of proteostasis, disabled macroautophagy, mitochondrial dysfunction, cellular senescence, stem cell exhaustion, altered intercellular communication, deregulated nutrient-sensing, chronic inflammation, and dysbiosis [132]. Chronic inflammation, also known as inflammaging and immunosenescence, is a consistent feature of aging and age-related diseases [69]. This raises the notion that the translocation of gut microbiota and bile acid disorders may encourage age-related immune dysregulation.

Of some importance, rodents given LCA showed decreased lipid necrosis, better mitochondrial structure, limited reactive oxygen species production, and lived longer [70]. And, somewhat predictably, bile composition changes with age [71,72,73]. Centenarians (individuals at age > 100 years) had a distinct gut microbiome and unique secondary bile acids, including various isoforms of LCA. The bile acids and metabolites from such individuals were antimicrobial to Gram-positive (but not Gram-negative) multidrug-resistant bacteria [7,133,134].

Even though longevity is the people’s long-standing pursuit, to clarify the exact mechanism and pick out the ‘Mr. Key’ in this process is still the most important to success. Fortunately, many lines of evidence have clearly demonstrated that modifying the bile acids profile can delay aging effectively. A Mediterranean diet or calorie restriction altered the gut microbiota and bile acids, which was posited to improve health during aging [135,136,137,138]. In rodents, methionine restriction increased macroautophagy/autophagy and altered bile acid conjugation and levels to lengthen lifespan [139,140]. Modifying the bile acid profile with medications or fecal transplantation also relieved age-associated metabolic dysregulation in mice [141,142].

## 4. Conclusions and Perspective

A well-functioning intestinal barrier maintains normal function of the digestive tract and positively impacts the entire individual. This is not surprising as the intestinal barrier is the largest interface between the individual and the environment. It is sensitive to changes in many external and internal factors such as diet, pollution, alcohol, drugs, stress, and life cycle [143,144,145]. Bile acids are the only small molecules synthesized de novo and metabolized by the digestive system. Bile acids traverse the enterohepatic circulation 6–8 times per day and have a complex dynamic interaction with the gut microbiota [3]. It is reasonable to ascribe a central role to bile acids in intestinal homeostasis.

Given the physiological significance of bile acids signaling, greater insight into the complex relationship between bile acids and the intestinal barrier could uncover safe therapies for intestinal, hepatobiliary, and age-related diseases. Encouraging is the considerable progress achieved in modifying bile acid signaling through the use of bile acids and their derivatives (e.g., UDCA and OCA), targeting BARs, and through the regulation of the gut microbiota (e.g., by fecal microbiota transplant). However, considering interindividual variability in disease status and tolerance to treatment, personalized medicines are still in need. 

Bile acid metabolism is a multi-step physiological process, except for 12a-hydroxylated, many other modification processes, such as 7a-dehydroxylation, hydrolysis, and epimerization, may also play significant roles in the physiochemical and signaling properties of BAs. For example, some derivatives of LCA, formed by isomerization, can affect the function of the intestinal immune barrier through suppressing Th17 cell differentiation and increasing the differentiation of Treg [40,41]. Even the composition of BAs in mice differs from that of humans because the majority of CDCA in mice is typically converted to MCA [135]. But, according to the most recent research, both have been reported to improve the function of gut barrier [18,146,147].

In summary, fine-tuning the metabolism of bile acids is important in the homeostasis of the intestinal barrier. Disorders of bile acids and BAR signaling are involved in intestinal barrier dysfunction. Targeting bile acids and bile acid pathways may provide treatments to the related diseases arising from the deterioration of the intestine barrier. Still, further studies are warranted to elucidate the underlying mechanisms of action. Large, controlled, longitudinal clinical studies will assist in this.

## Figures and Tables

**Figure 1 cells-12-01888-f001:**
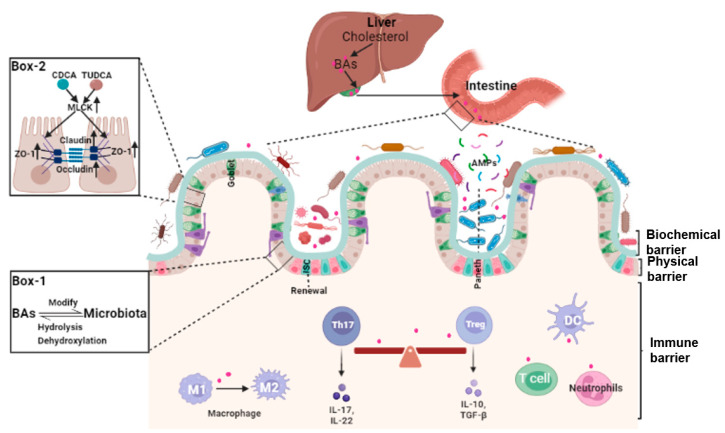
The roles of bile acids in the homeostasis of the intestinal barrier. Bile acids (BAs) are synthesized in the liver, stored in the gallbladder, and secreted into the intestine. In the gut, bile acids modify the growth of gut microbiota. Reciprocally, bile acids are dehydroxylated and/or de-conjugated by the gut microbiota to form secondary or tertiary bile acids (Box 1). Bile acids such as CDCA and TUDCA are involved in the maintenance of the integrity of the intestinal barrier through affecting the expression of tight junction proteins (Box 2). Bile acids are also involved in modifying the gut microbiota and intestinal mucosal lamina propria local immune system. In this location, they regulate macrophage polarization, inflammatory T helper 17 (Th17) cells and regulatory T cell (Treg) cells, and dendritic cells (DCs).

**Figure 2 cells-12-01888-f002:**
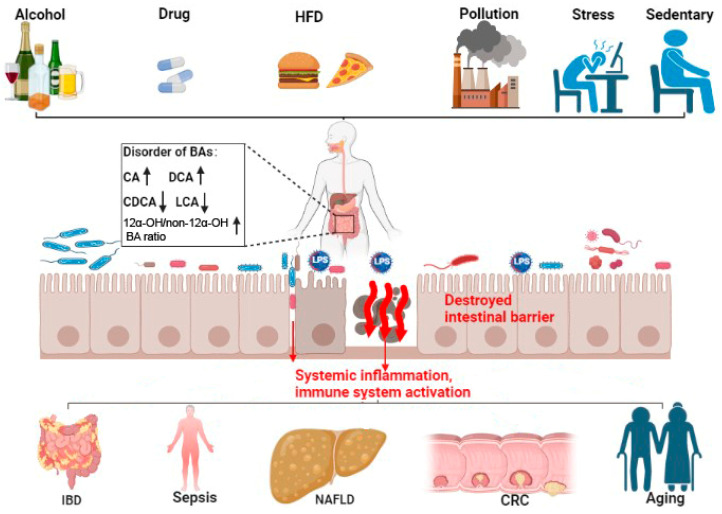
Bile acid disorder and intestinal-barrier-dysfunction-related diseases. Bile acid (BA) metabolic disorder can be caused by alcohol intake, drugs, HFD, pollution, stress, and a sedentary lifestyle. Bile acid metabolic disorders can cause increased CA and DCA (shown by ↑), decreased CDCA and LCA (shown by ↓), and increased 12α-OH/non-12α-OH BAratio (↑). This imbalance in the BA profile can damage the intestinal barrier, increase the translocation of pathogenic microbiota and metabolites (shown by red arrows), and promote systemic inflammation and immune system activation. These then potentiate IBD, sepsis, NAFLD, CRC, and aging. HFD, high fat diet; LPS, lipopolysaccharides; CA, cholic acid; CDCA, chenodeoxycholic acid; DCA, deoxycholic acid; LCA, lithocholic acid; FXR, farnesoid X receptor; TGR5, Takeda G protein-coupled receptor 5; IBD, intestinal inflammatory diseases; NAFLD, non-alcoholic fatty liver disease; CRC, colorectal cancer.

**Table 1 cells-12-01888-t001:** Bile acids and intestinal-barrier-dysfunction-related diseases.

Diseases	BAs	BARs	Mechanism	Reference
IBD	PBAs↑ SBAs↓	Inhibit FXR and TGR5	Alter the expression of tight junction proteins and the renewal of intestinal stem cells; inhibit Paneth cell function and type I interferon signaling.	[32,47,48,49,50,51]
Sepsis	TBAs↑ TωMCA↑	Inhibit FXR and TGR5	TUDCA prevents sepsis through inhibiting TGR5-NF-κB and endoplasmic reticulum stress. TBAs increase the gut barrier integrity.	[52,53,54,55,56,57]
NAFLD	DCA↑ Conjugated PBAs↑ Unconjugated PBAs↓	Inhibit FXR	Interfere with TLR4/TGF-β1 signaling pathway, activate autophagy and intestinal integrity, decrease intestinal barrier function, and induce changes in microbiota composition.	[58,59,60,61,62,63,64]
CRC	CA↑ DCA↑ CDCA↓ LCA↓	Inhibit FXR-FGF15 Enhance TGR5	Promotes dysregulated immunity, loss of the intestinal barrier, invasion of microbial and pathogenic metabolites, and increases inflammation.	[65,66,67,68]
Aging	LCA↓ iso-LCA↓ 3-oxo-LCA↓	Inhibit RORγt	Encourage age-related immune dysregulation and promote chronic inflammation.	[42,69,70,71,72,73]

Abbreviations: BAs, bile acids; PBAs, primary bile acids; BARs, bile acids receptors; TGR5, Takeda G protein-coupled receptor 5; FXR, farnesoid X receptor. TωMCA, tauro-ω-muricholic acid; CA, cholic acid; DCA, deoxycholic acid; LCA, lithocholic acid; UDCA, ursodeoxycholic acid. ↑ means increased level of bile acids, and ↓ means decreased level of bile acids.

## Data Availability

Not applicable.

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
