# Peer review of "Bile Acids, Intestinal Barrier Dysfunction, and Related Diseases"

_cells, 2023, doi:10.3390/cells12141888_

Round 1
Reviewer 1 Report
The review entitled “Bile acids, intestinal barrier dysfunction and related diseases" by Shi, L. et al. highlights the crucial role of bile acids in maintaining the integrity and function of the intestinal barrier.
This review discusses the influence of bile acids on the permeability of intestinal mucosa by regulating the expression of tight junction proteins, the role of bile acids on intestinal epithelial and immune cell homeostasis. Bile acid-related intestinal barrier dysfunction contributes to the development and progression of inflammatory bowel disease as well as some other disease such as NAFLD, colorectal cancer and aging.
A typo remains - "." line 150
The review was well-written and comprehensively described. The third part of the article would benefit from a table, classified by pathology and summarizing all the bile acids involved and associated with other markers and/or receptors.
Author Response
Thank reviewer for the positive comments. We have addressed the comments accordingly.
- The typo is corrected.
- As suggested, a table to summarize bile acids, pathology, receptors and reference numbers is added in the revised manuscript.
Reviewer 2 Report
This is a well written review of important topics in the bile acid research field. I only have one minor comments for the authors to consider.
1. The authors state that bile acid can be grouped in 12a-hydroxylated and non-12a-hydroxylated bile acids. However, I am not sure if I can agree with the authors on that this clearly contributes to the differential physiochemical and signaling properties of CA, DCA, LCA and UDCA as discussed. 7a dehydroxylation or epimerization probably plays significant roles, instead. In addition, changes in bile acid 12 hydroxylation probably has a more significant effect in mice than in humans due to the presence of hydrophilic MCAs in mice but hydrophobic CDCA in humans. Please discuss.
1. There are still some typos and grammar errors. please do a careful proof read.
2. references are missing on a few places.
Author Response
Thank reviewer for the insightful comments. We have revised the manuscript accordingly.
- A new paragraph is added in the discussion to highlight 7a- dehydroxylation or epimerization of bile acids, as well as the difference of bile acid species between mice and human.
- Typos and grammar errors are corrected accordingly.
- The missing references are added.